# SW16-7, a Novel *Ackermannviridae* Bacteriophage with Highly Effective Lytic Activity Targets *Salmonella enterica* Serovar *Weltevreden*

**DOI:** 10.3390/microorganisms11082090

**Published:** 2023-08-15

**Authors:** Jialiang Xu, Jia Li, Yi Yan, Pengjun Han, Yigang Tong, Xu Li

**Affiliations:** 1China Food Flavor and Nutrition Health Innovation Center, Beijing Technology and Business University, Beijing 100048, China; xujialiang@btbu.edu.cn (J.X.); eric_li07@163.com (J.L.); yanyi@btbu.edu.cn (Y.Y.); 2College of Life Science and Technology, Beijing University of Chemical Technology, Beijing 100029, China; lyhpj88@163.com (P.H.); tongyigang@mail.buct.edu.cn (Y.T.)

**Keywords:** bacteriophage, *Salmonella enterica*, *Ackermannviridae*

## Abstract

*Salmonella enterica* serovar *Weltevreden* is a foodborne pathogen commonly transmitted through fresh vegetables and seafood. In this study, a lytic phage, SW16-7, was isolated from medical sewage, demonstrating high infectivity against *S. Weltevreden*, *S. London*, *S. Meleagridis*, and *S. Give* of Group O:3. In vitro inhibition assays revealed its effective antibacterial effect for up to 12 h. Moreover, analysis using the Comprehensive Antibiotic Resistance Database (CARD) and the Virulence Factor Database (VFDB) showed that SW16-7’s genome does not contain any virulence factors or antibiotic resistance genes, indicating its potential as a promising biocontrol agent against *S. Weltevreden*. Additionally, a TSP gene cluster was identified in SW16-7’s genome, with TSP1 and TSP2 showing a high similarity to lysogenic phages ε15 and ε34, respectively, in the C-terminal region. The whole-genome phylogenetic analysis classified SW16-7 within the *Ackermannviridae* family and indicated a close relationship with *Agtrevirus*, which is consistent with the ANI results.

## 1. Introduction

*Salmonella enterica* serovar *Weltevreden* is a pathogen responsible for zoonotic disease, originating from India [1]. This serovar primarily causes self-limiting gastroenteritis with symptoms such as diarrhea, fever, and vomiting [2], and is prevalent in Southeast Asia [1,3,4,5]. Contaminated vegetables [6,7] and seafood are the primary transmission sources [8], and imported freshwater fish and marine shrimp from Southeast Asia are likely to be associated with infections caused by *S. Weltevreden* in North America [9,10,11]. In addition, this serovar has been detected in broilers [12] and pigs [13], and its presence in a variety of food matrices is likely to contribute to its global spread. Over the past two decades, there has been a significant increase in the number of outbreaks caused by *S. Weltevreden* in various regions [14,15,16,17,18]. Severe outbreaks have been reported, such as in India, where 150 students aged 20 to 30 were affected by acute watery diarrhea caused by *S. Weltevreden* infection [19], and in China, where 40 cases of diarrhea were reported due to *S. Weltevreden* infection in the southern coastal region [20].

It should be noted that while *S. Weltevreden* strains isolated from humans have not shown significant antimicrobial resistance, strains found in animals or food have demonstrated severe antimicrobial resistance (AMR) phenotypes, including resistance to aminoglycosides, tetracycline, nalidixic acid, ampicillin, and sulfonamide [21]. These multidrug-resistant strains of *S. Weltevreden* pose a potential challenge to public health and food safety, highlighting the need for control measures to prevent their spread.

Phages are being increasingly recognized as a green and environmentally friendly antibacterial agent for food applications [22,23], with several phage preparations, including SalmoFresh™ and SalmoLyse^®^ [24], now approved as biocontrol agents of *Salmonella* spp. Unlike chemical and thermal sterilization methods, phages do not alter the flavor and texture of food, making them more suitable for raw foods such as fruit salads, sashimi or sushi that are more likely to be contaminated with *S. Weltevreden*. Moreover, phages can act as antibacterial agents at all stages of the food processing chain, including animal feed, primary production, household, or food service establishments [25].

Numerous studies have focused on the use of phages to control *S. Typhimurium*, *S. Enteritidis* and *S. Derby* [26,27], but these phages have shown limited activity against *S. Weltevreden*. Given the public health significance of *S. Weltevreden*, this study isolated and characterized a novel *Ackermannviridae* phage, designated SW16-7, which targets *S. Weltevreden*. SW16-7 demonstrated high infectivity against *S. Weltevreden* and other *Salmonella* serovars belonging to Group O:3, together with excellent heat stability and pH tolerance. These findings suggest that SW16-7 may be a potential candidate for controlling the spread of *S. Weltevreden*.

## 2. Materials and Methods

### 2.1. Bacteria Strains and Growth Conditions

Bacterial strains used in the study were sourced from the Chinese Centre for Disease Control and Prevention (China CDC) and confirmed using standard biochemical tests and a Salmonella diagnostic antisera kit (SSI Diagnostica, Shanghai, China). The strains were stored at −80 °C with 15% glycerol until use. Prior to experiments, the bacteria were incubated in Luria-Bertani (LB) broth (Oxoid) at 37 °C for 12 h. For agar medium, 1.5% agar (Luqiao, Beijing, China) was added as required.

### 2.2. Phage Isolation and Purification

To isolate the phage, an enrichment method [28] was employed using medical sewage samples collected from a hospital in Beijing, China. Firstly, debris was removed from 50 mL of sewage through centrifugation at 8000× *g* for 20 min, and the resulting supernatant was filtered through a 0.22 μm filter (Millipore, Beijing, China) to obtain the phage lysate. Next, 4 mL of the log-phase host bacteria (*S. Weltevreden* strain SA161) were mixed with the phage lysate and incubated at 37 °C for 24 h. The mixture was then centrifuged at 8000× *g* for 15 min, and the supernatant was filtered through a 0.22 μm filter to check for phage activity using the spot test and double-agar overlay method. Following the identification of plaques, the phage was purified through three consecutive purification rounds, resulting in the isolation of the final bacteriophage, designated as SW16-7.

### 2.3. Transmission Electron Microscopy

SW16-7 was visualized using transmission electron microscopy (TEM). Approximately 20 μL of the sample, with a titer of around 10^10^ PFU/mL, was dispensed onto parafilm. A grid with a carbon-supported formvar film was then floated on the phage suspension for 1 min, followed by staining with sodium phosphotungstate (pH 6.8) for 1 min. The 400-mesh grid (Sigma-Aldrich, Beijing, China) was subjected to UV radiation for 10 min to inactivate the phage. Images were acquired using a TECNAI 12 (FEI, Hillsboro, Oregon, USA) equipped with a 16 MegaPixel TEM CCD Camera Morada^G3^ (EMSIS, Muenster, Germany).

### 2.4. Determination of Host Ranges

The lytic activity of SW16-7 was assessed against 11 strains of *S. Weltevreden* and 40 strains of *S. London*, *S. Meleagridis*, *S. Give* and *S. Muenster* from groups O:3, which have similar O-antigen structures. Additionally, 133 strains from other serotypes, including Group O:4, Group O:9, Group O:7, Groups O:6,8, Groups O:8, Group O:9,46, and Group O:17, were tested to determine the phage’s host range.

The host range analysis was conducted using spot tests, where the tested strains were mixed with 0.5% LB agar as the overlay, while 1.5% LB agar served as the bottom layer. Phage lysates with a concentration of 1.0 × 10^6^ PFU/mL were spotted onto the plates, which were then incubated at 37 °C for 16 h. The sensitivity of the test bacteria was determined by observing the clarity of the spots after incubation. All experiments were performed in triplicate to ensure accuracy and reproducibility.

### 2.5. Thermal and pH Stability Test

The stability of SW16-7 was assessed under different thermal and pH conditions. To evaluate pH stability, a phage lysate was mixed with 1 mL of LB adjusted to various pH levels ranging from 2.0 to 14.0. The mixtures were incubated at 37 °C for 1 h, followed by determination of the phage titer using dilution and double-layer plating.

For thermal stability assessment, phage suspensions were exposed to different temperatures (40 °C, 50 °C, 60 °C, 70 °C, 80 °C, and 90 °C) for 1 h. After incubation, the phage titer was determined by dilution and plating.

### 2.6. Optimum Multiplicity of Infection (MOI)

To determine the optimal multiplicity of infection (MOI) for SW16-7, SA161 were cultured with phage lysates at varying ratios of phage concentration ranging from 10^−6^ to 10. The titer was then measured after each of the three independent experiments performed for each MOI. The MOI with the highest titer was identified as the optimal MOI for SW16-7.

### 2.7. One-Step Growth Curve

To determine the replication characteristics of SW16-7, a one-step growth curve was performed. The host strain was cultured in LB until it reached the exponential phase (3 × 10^8^ CFU/mL) after inoculation into fresh LB at a ratio of 1%. SW16-7 (1 × 10^7^ PFU/mL) and SA161 (1 × 10^8^ CFU/mL) were mixed with 6 mL of LB broth at an optimal MOI of 0.1 and incubated at 37 °C. Samples of 100 μL were collected from each mixture at 10-min intervals for the first 60, 90, and 120 min, diluted, and plated on double-layer plates to determine the phage titers. The burst size was calculated by dividing the maximum phage titer at the plateau phase by the initial infective titer of the infection. The experiments were conducted in triplicate, and the results were analyzed.

### 2.8. In Vitro Inhibitory Assessment of Phage SW16-7

The inhibition of SW16-7 was assessed through two in vitro experiments. In the first experiment, the host bacterium SA161 was incubated in LB medium until it reached an optical density of approximately OD600 = 0.5, corresponding to a bacterial concentration of around 10^8^ CFU/mL. The treatment group was mixed with 200 μL of SA161 and SW16-7 at an MOI of 0.1, while the control group received 200 μL of SA161 and an equal volume of LB liquid as a substitute for SW16-7. The growth of host bacteria in both groups was monitored every hour for a duration of 24 h using Bioscreen C (Oy Growth Curves Ab Ltd., Helsinki, Finland).

The protocol of the second experiment was similar to the first experiment. In brief, both treatment and control groups were cultured to the exponential phase with 20 mL each. The treatment group received phage SW16-7 at an MOI of 0.1, while the control group received an equal volume of LB liquid. Colony counting of SA161 was recorded at 4-h intervals over a period of 24 h using the plate count method.

### 2.9. Genome Sequencing and Bioinformatics Analysis

Phage DNA was extracted and purified using a Lambda Phage Genome Purification Kit (ABigen, Beijing, China). The quality of the raw sequencing reads was assessed using FastQC software version 0.11.9 (http://www.bioinformatics.babraham.ac.uk/projects/fastqc/). High-quality reads were assembled into contigs using the SPAdes Assembler version 3.13.0. The genome of SW16-7 was then submitted to the RAST server for CDS prediction and annotation [29]. The functions of the predicted CDSs were verified using BLASTP. Conserved domains of CDSs were identified by searching the PFAM database [30]. The presence of tRNA genes in the genomic sequences was determined using a tRNAscan-SE web server [31]. To identify CDSs associated with antibiotic resistance genes (ARGs) and virulence factors, the Comprehensive Antibiotic Resistance Database (CARD) [32] and the Virulence Factor Database (VFDB) [33] were searched. A circular map of the SW16-7 genome was generated using Proksee (https://proksee.ca/). The protein domain structures were modeled using IBS2.0 web server [34].

The phage genomes and their corresponding classifications used for phylogenetic analysis were obtained from the National Center for Biotechnology Information (NCBI) database. The core genes from all phage genomes were concatenated and aligned, and a maximum likelihood tree was constructed using IQ-TREE (version 1.6.12) with 1000 bootstrap replications. The general time reversible (GTR) model was applied during the phylogenetic analysis. A pairwise average of the nucleotide identities (ANIs) among the 31 genomes included in this study was computed using an OrthoANI tool [35].

For the phylogenetic analysis based on a terminase large subunit, the sequences were aligned using a MAFFT plugin, and the tree was generated using a RAxML plugin within the Geneious Prime software (version 9.0.2). A tvBOT online service was utilized for visualizing all the generated trees [36].

### 2.10. Statistical Analysis

The experimental designs for the one-step growth assay, TEM analysis, stability analysis, and in vitro inhibition assessment using the Bioscreen C and the plate count method were conducted with three biological replicates. For the in vitro inhibition assessment using the plate count method, a two-tailed unpaired Student’s t-test was conducted to compare the control group and the experimental group at each time point (0, 4, 8, 12, 16, 20, 24). The statistical analysis was conducted using SPSS software (SPSS 19.0; SPSS). The data obtained from these experiments were expressed as mean ± standard deviation (SD). Graphs illustrating the results were generated using GraphPad Prism version 8 (version 8.3.0) software.

## 3. Results

### 3.1. Morphology of Phage SW16-7

The bacteriophage SW16-7 produced small, clear plaques measuring 1–2 mm in diameter on a lawn of host bacteria, as shown at the top of Figure 1. To further characterize SW16-7, TEM was used to study its morphology. The results revealed that SW16-7 has a typical regular polyhedral shape, with a head diameter of 100 ± 2.5 nm, a contractile tail length of 110 nm ± 10 nm, and a width of 18 ± 2.3 nm, as depicted in Figure 1. A total of 20 bacteriophages was measured, providing the basis for calculating the mean and standard deviation.

### 3.2. Biological Features of Phage SW16-7

Phage SW16-7 exhibited high efficacy against the host at all tested MOIs, with an optimal multiplicity of infection of 0.1, resulting in a titer of up to 10^11^ PFU/mL (Figure 2A). The one-step growth curve indicated that the latent period was approximately 10 min, and the average phage burst size was 57.1 ± 9 PFU/cell under optimal MOI conditions (Figure 2B). The phage’s stability under different thermal and pH conditions was evaluated by measuring plaque-forming units. The results showed that the lytic activity of SW16-7 remained stable when incubated between pH 4 and pH 12, but outside of that range, the activity decreased (Figure 2C). In the thermal stability test, the phages remained stable between 40 and 60 °C, but a gradual decrease in activity was observed between 60 and 80 °C, and complete inactivation was achieved at 90 °C (Figure 2D).

Figure 3 demonstrates the inhibitory effect of phage SW16-7 against the host bacteria SA161 over time. The results revealed a significant inhibitory effect of SW16-7 against host bacteria growth from the 8–20 h, resulting in a maximum reduction of approximately three orders of magnitude (Figure 3B). Although the inhibitory capacity of SW16-7 decreased after 12 h, it continued to inhibit host bacteria growth even at 20 h. This sustained inhibitory effect between 4 and 24 h was also observed in the OD600 curve (Figure 3A), indicating the continuous inhibition of SW16-7 against SA161.

### 3.3. General Features of the Phage SW16-7 Genome

Phage SW16-7 has a 155,665 bp double-stranded DNA genome with a GC content of 50.1%. The genome map in Figure 4 shows coding domain sequences (CDSs) colored by their functional categories. The RAST server annotated 205 CDSs, of which 71 were predicted as functional proteins and 129 as hypothetical proteins. Among the hypothetical protein genes, 77 (59.69%) were on the negative strand and 52 (40.31%) were on the forward strand, with no homologs found in the non-redundant database by BLASTP analysis. Most functional protein genes (51 of 71; 71.83%) were on the negative strand. The tRNAscan-SE identified five tRNA genes on the forward strand. The SW16-7 genome does not contain any virulence factors or antibiotic resistance genes. Appendix A summarizes the general features of the predicted CDSs in the SW16-7 genome.

### 3.4. Specific Features of the Phage SW16-7 Genome

The functional proteins encoded by the SW16-7 genome have been classified into five groups based on their annotations, including Morphogenesis, Auxiliary metabolism, DNA metabolism, Gene expression, and Lysis.

#### 3.4.1. Morphogenesis-Related Genes

The genome of the SW16-7 phage contains 30 putative morphogenesis-related proteins, which likely play a role in the assembly of various phage structures, such as the head, tube, tails, tail fiber, and baseplate. Among these, 16 proteins were classified as T4-like proteins, indicating a potential evolutionary relationship between SW16-7 and the T4 phage. 

Tail fiber (TF) and tail spike (TSP) are essential components of tailed phages for host recognition [37,38,39,40,41]. Only one TF gene (CDS33) was identified on the SW16-7 genome, and it exhibited 99.89% protein identity to the *Salmonella* phage P46FS4. VriC, a highly conserved protein that serves as a marker for the TSP gene cluster in the Ackermannviridae family, follows a typical arrangement of a VriC-TSP(n)-Hypothetical protein-Hypothetical protein-Baseplate wedge [42]. In SW16-7, CDS187 was identified as VriC, CDS192 as a basal wedge, and CDS191 and CDS192 as hypothetical proteins through BLASTP analysis. Additionally, CDS188 and CDS189 were annotated as two types of TSPs. Consequently, the TSP gene cluster in SW16-7 consisted of six genes, as depicted in Figure 5A.

The domain architecture of TSP1 and TSP2 is illustrated in Figure 5B. TSPs are typically divided into N-terminal and C-terminal regions based on their functional domains. The N-terminal region is involved in base wedge assembly, while the C-terminal region exhibits catalytic enzyme activity. Referring to the structure of TSP in CBA120, we found that both TSP1 and TSP2 have a gp10-like module at the N-terminus. The gp10 module is composed of XD domains (XD1, XD2, XD3), and the number and combination of XD domains vary among different TSPs [43]. In SW16-7, TSP1 residues 1–177 showed 75% homology with TSP2 of the CBA120 phage, corresponding to the XD2 and XD3 sequences, thus making up its gp10 module. On the other hand, TSP2 residues 1–250 showed 71% homology with TSP4 of the CBA120 phage, corresponding to the XD1, XD2, and XD3 sequences, thus making up its gp10 module. After the gp10-like module, both TSPs share the Tail_spike_N domain (PF18668.4), which marks the boundary for the N-terminal of several TSPs. Therefore, the sequence after the Tail_spike_N domain is annotated as the C-terminal region. 

PFAM analysis did not reveal any conserved domains in the C-terminal region of TSP1, while TSP2 has an additional Beta_helix domain (PF13229.9) between amino acid positions 487–681, similar to a pectinase. BLASTN analysis showed that the C-terminal sequence of TSP2 has 94.27% homology with the TSP of ε15 phage (231–1069aa), which has endorhamnosidase activity capable of degrading Group O:3,10 *S. enterica* O-polysaccharide polymers. Conversely, the C-terminal sequence of TSP1 has 94.27% homology with the TSP of ε34 phage (114–666aa), which exhibits enzymatic hydrolysis activity. Appendix A show the protein sequence comparison results between TSP1 and the TSP of the ε34 phage and between TSP2 and the TSP of the ε15 phage, respectively.

#### 3.4.2. DNA Metabolism-Related Genes

The SW16-7 genome contains a DNA metabolism module, which is composed of various genes that play crucial roles in fundamental processes such as DNA replication, repair, and recombination. These processes are essential for the efficient transfer of genetic information during the viral life cycle. The DNA metabolism module of SW16-7 includes genes for DNA replication, such as DNA polymerase (CDS7), DNA polymerase clamp-related proteins (CDS148, CDS149, CDS151), DNA topoisomerases (CDS39, CDS40), DNA helicase (CDS125), helicase loader (CDS62), DNA primase (CDS75, CDS108), DNA binding protein (CDS35, CDS88, CDS126), and ribonucleotide reductase (CDS102, CDS103).

Furthermore, the SW16-7 phage also encodes genes for DNA repair and recombination, including homing endonuclease (CDS63), DNA ligase (CDS67), and UvsW (CDS153) for DNA repair, and UvsX (CDS77), UvsY (CDS156), and recombination-related endonuclease (CDS120) for DNA recombination. Notably, these genes account for approximately 29.57% (21/71) of the functional genes annotated in the SW16-7 genome, emphasizing their significance in genetic information transfer and virion assembly.

Certain *Salmonella* phages possess the ability to undergo thymidine modifications, resulting in hyper-modification of their DNA. This serves as a protective mechanism against cleavage by host restriction endonucleases [44]. Upon examination of the SW16-7 genome, as per the PFAM database, we discerned five CDSs (CDS81, CD145, CD15, CDS82, CDS11) that could potentially participate in thymidine modifications. CDS15, CDS82, and CDS11 exhibit protein similarities of 87.51%, 87.72%, and 82.30%, respectively, with the aGPT-Pplase2, P-loop kinase-1, and P-loop kinase-2 of *Salmonella* phage ViI. CDS145 shows a protein similarity of 94.28% with the aGPT-Pplase1 of *Salmonella* phage P46FS4, and CDS81 shared a 97.10% similarity with the thymidylate synthase of *Salmonella* phage SKML-39. 

#### 3.4.3. Gene Expression-Related Genes

This group of genes is involved in various aspects of gene expression regulation. CDS90 encodes a T4-like late promoter transcription accessory protein, which is involved in enhancing the transcription of late genes. CDS91, which belongs to the FmdB family, encodes a transcriptional regulator that assists RNA polymerases in transcription, indicating its potential role in gene expression control. CDS123 encodes Ribonuclease HI, an enzyme involved in the degradation of RNA-DNA hybrids, which are formed during DNA replication, recombination, and repair. Finally, CDS147 encodes an endoribonuclease that acts as a translational repressor of early genes, suggesting its role in post-transcriptional regulation. 

#### 3.4.4. Auxiliary Metabolism-Related Genes

Bacteriophages often carry auxiliary metabolic-related genes that regulate the host cell metabolism for efficient phage replication. The group includes 11 genes, such as CD45 (Tk.4 protein), CD79 (dUTP diphosphatase), CD104 (phosphate starvation-inducible protein PhoH), CD121 (cell division trigger factor), CD139 (nicotinamide phosphoribosyltransferase), CD46 (DexA exonuclease), CD140 (ribose-phosphate pyrophosphokinase), CD51 (serine/threonine protein phosphatase), CD81 (thymidylate), CD101 (glutaredoxin), and CD55 (deoxycytidylate deaminase).

#### 3.4.5. Lysis-Related Genes

Endolysins are phage-encoded enzymes that degrade bacterial cell walls. In the SW16-7 genome, the endolysin is encoded by the gene CDS105. The endolysin is a relatively small protein with a molecular mass of 28.6 kDa and a theoretical pI of 9.05.

The SW16-7 endolysin has a modular structure identified by PFAM analysis, consisting of a PG_binding_1 domain (9–64 aa) known as a CBD, and a muramidase domain (90–264 aa) known as an EAD, located at the N-terminal and C-terminal regions, respectively. The CBD binds specifically to peptidoglycan, a major component of the bacterial cell wall, while the EAD cleaves the glycosidic bond between the N-acetylmuramic acid and N-acetylglucosamine residues in the peptidoglycan backbone.

### 3.5. Phylogenetic Analysis of Phage SW16-7

A total of 54 phage genomes was identified through BLASTN analysis, exhibiting over 80% identity and over 60% coverage with the genome of SW16-7. These phage genomes taxonomically belong to three genera within the Ackermannviridae family: *Agtrevirus*, *Limestonevirus*, and *Kuttervirus*. To further analyze their relationships, phylogenetic analysis was conducted using the core genes of the 54 identified phage genomes.

The whole-genome phylogenetic tree revealed the presence of three distinct clusters: cluster I consisting of 34 *Kuttervirus* phages, cluster II comprising 15 *Limestonevirus* phages, and cluster III containing 5 *Agtrevirus* phages (Figure 6). Notably, SW16-7 showed closer association with phages such as *Shigella* phage MK-13, *Salmonella* phage SKML-39, *Salmonella* phage P46FS4, and *Shigella* phage Ag3 within cluster III. The results obtained from the terminase large subunit analysis were consistent with those obtained from the comprehensive analysis of core genes across the genomes (Figure 7). 

ANI values were calculated to assess nucleotide identity using whole-genome sequences. The ANI values between SW16-7 and the phages in cluster III were remarkably high (>95.57%), while the genomes of closely related phages had ANI values ranging from <88.24% to >74.47% (Table 1). These ANI results were consistent with the phylogenetic analysis, confirming the assignment of phage SW16-7 to the *Agtrevirus* genus.

### 3.6. Host Range of Phage SW16-7

Table 2 presents the lysis results of the SW16-7 phage against different serovars of *Salmonella* and *Shigella*, including the number of tested strains and the proportion of lysis isolates for each serovar. The study found that among the *Salmonella* serovars tested, *S. Weltevreden* and *S. Muenster* had a 100% lysis rate, while *S. London*, *S. Meleagridis*, and *S. Give* had high lysis rates ranging from 81.81% to 85.71%. On the other hand, *S. Derby*, *S. Stanley*, *S. Typhimurium*, *S. Rissen*, *S. Corvallis*, and *S. Kentucky* had low lysis rates ranging from 6.25% to 11.76%, while *S. Enteritidis* had no lysis isolates.

The *Salmonella* serovars were grouped based on their O antigens [45], and it was found that the high lysis rates were observed in serovars belonging to groups O:3,10 and O:6,8. Meanwhile, the low lysis rates were mainly observed in serovars belonging to groups O:4 and O:8. Additionally, *S. Hillingdon*, *S. Gateshead*, and *S. Michigan* could also be lysed by SW16-7, belonged to groups O:9,46 and O:17, respectively.

## 4. Discussion

Previous studies have focused on phage typing of *S. Weltevreden*, which have been widely adopted and proven effective in the epidemiological investigation [46,47,48]. However, the unavailability of complete genome sequences of the lytic phages employed in these schemes currently limits their direct applicability for the biocontrol of *S. Weltevreden*.

SW16-7 can be generally classified as an aquatic phage due to its ability to be transmitted through aquatic products and survive in hydroacoustic environments. Aquatic phages have been identified as reservoirs of ARGs, contributing to the dissemination of antibiotic resistance in the environment [49,50]. Nevertheless, SW16-7 stands out as it lacks any ARGs or virulence factor genes in its genome, ensuring its safety as an antimicrobial agent from a genomic perspective. The consumption of raw seafood, an important source of *S. Weltevreden* infection in humans, often involves the presence of seawater on the seafood’s surface during transportation. To effectively control *S. Weltevreden* in seafood using phages, it is crucial for the phages to adapt to the relatively high pH range of seawater, typically between 8.0 and 8.25 [51]. Our investigations have demonstrated that SW16-7 exhibits robust activity within this pH range, making it a potential candidate for controlling *S. Weltevreden* contamination in seafood. Considering its genomic safety and pH tolerance, SW16-7 holds great potential for the biological control of *S. Weltevreden* contamination.

However, when used alone, SW16-7 may not yield satisfactory long-term inhibition, as indicated by the results depicted in Figure 3 (12 to 24 h). This reduced efficacy over extended periods could be attributed to the emergence of phage-resistant bacterial strains [52,53]. To counteract the development of phage resistance, the use of phage cocktails has proven effective and cost-efficient [54,55,56]. Therefore, it is anticipated that SW16-7 will be incorporated as part of a phage cocktail in conjunction with other phages in future applications.

In addition to its specific lytic activity against *S. Weltevreden*, phage SW16-7 has shown promising potential in lysing other Salmonella strains belonging to Group O:3 (formerly group E), including *S. Muenster*, *S. London*, *S. Anatum*, *S. Give*, and *S. Meleagridis*. This indicates that SW16-7 can be categorized as an O:3 Group-specific phage. TSPs are receptor-binding proteins found in *Ackermannviridae* phages, playing a crucial role in determining the host range [42]. Typically, these phages possess three or four distinct TSPs [57,58], enabling them to infect various Gram-negative bacteria [43,59,60]. The C-terminal regions of TSPs facilitate the degradation of lipopolysaccharides (LPS) on the host bacteria’s surface, allowing the phage to access the outer membrane and initiate infection [61,62,63]. The literature suggests that the C-terminal region can be exchanged within a genus or acquired from distantly related phages, influencing phage host recognition [41,61]. Our investigation identified two types of TSPs in SW16-7 phages, specifically the C-terminal regions of TSP1 and TSP2 derived from lysogenic phages ε15 and ε34, respectively. These TSPs possess endorhamnosidase and endogalactosidase activities [64,65], enabling them to selectively target strains belonging to the Group O:3,10. Interestingly, strains within Group O:9,46 also possess factors O:3 and O:10, with the latter being less prominent [45]. This may explain why SW16-7 can effectively lyse not only O:3 group strains but also *S. Hillingdon* and *S. Gateshead*, both belonging to the Group O:9,46. However, the limited lysis observed in Group O:4, Group O:7, Groups O:6,8, and Group O:17 remains poorly understood and could be influenced by strain-specific factors.

The construction of phage cocktails comprising phages from diverse genetic backgrounds has proven effective in minimizing the emergence of phage resistance. However, existing phage cocktails targeting *Salmonella*, such as *S. Enteritidis*, *S. Newport*, and *S. Dublin*, have demonstrated limited efficacy against Group O:3,10 strains [66,67,68]. In light of this, we propose that the inclusion of SW16-7 in these cocktails could expand their host range and introduce novel receptor binding sites, thereby potentially delaying the development of phage resistance. This strategic addition of SW16-7 to the phage cocktail formulation has the potential to enhance its effectiveness in combating Group O:3,10 strains of *Salmonella*.

## Figures and Tables

**Figure 1 microorganisms-11-02090-f001:**
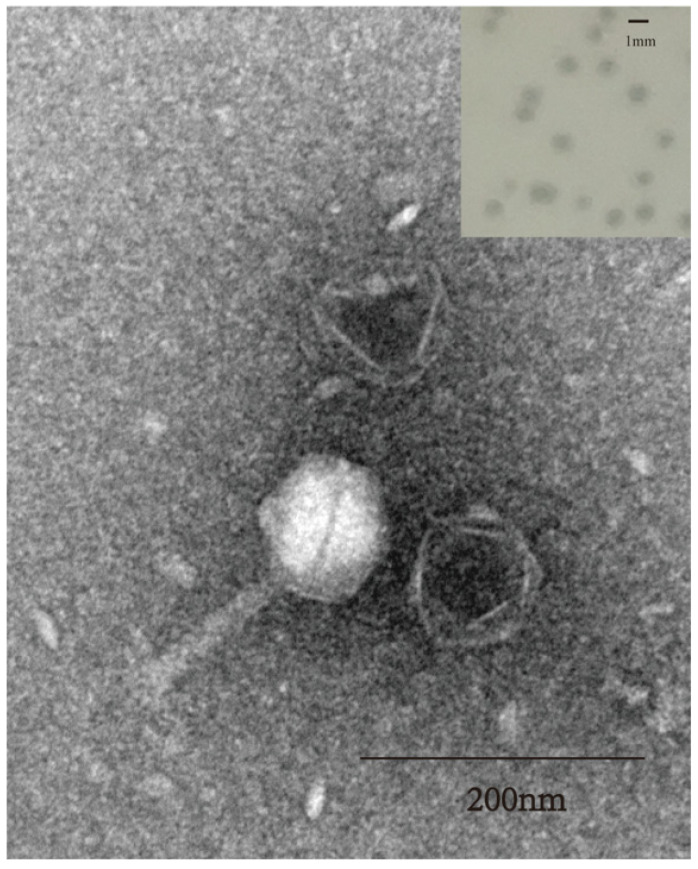
Morphological properties of phage SW16-7. Transmission electron micrograph of SW16-7. Scale bar indicates 200 nm; The upper right figure shows plaques formed by SW16-7 on the SA161 lawn using the double agar overlay method, with incubation at 37 °C for 12 h. Scale bar indicates 1 mm.

**Figure 2 microorganisms-11-02090-f002:**
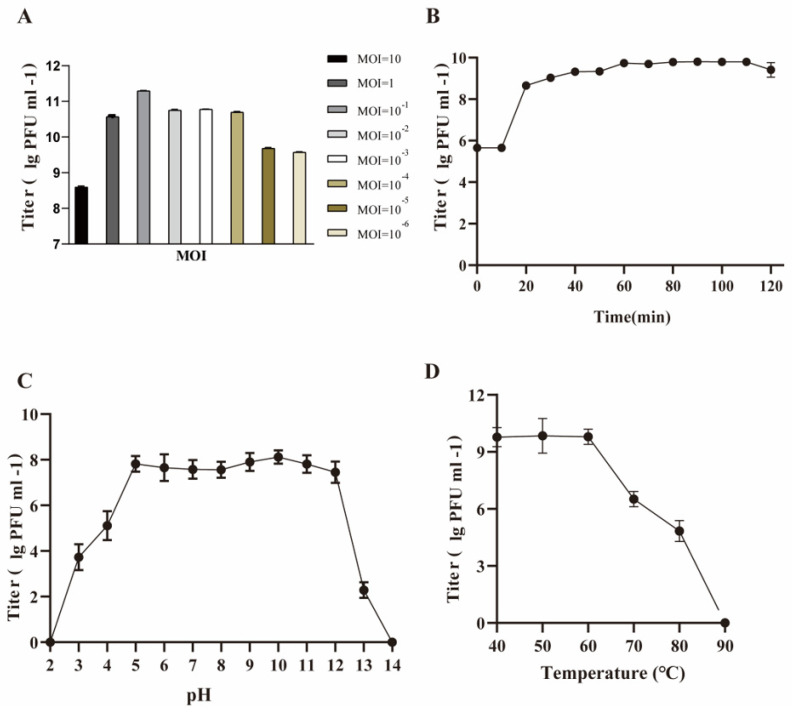
Biological properties of phage SW16-7. (**A**) The multiplicity of infection of phage SW16-7. Error bars, s.d. (n = 3 biological replicate) (**B**) One-step growth curve of phage SW16-7. Error bars, s.d. (n = 3 biological replicate) (**C**) Tolerance of the phage SW16-7 to different pH treatment. Error bars, s.d. (n = 3 biological replicate) (**D**) Thermal tolerance of the phage SW16-7. Error bars, s.d. (n = 3 biological replicate).

**Figure 3 microorganisms-11-02090-f003:**
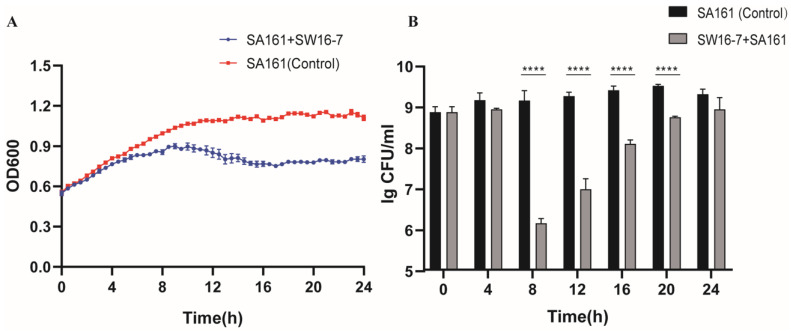
In vitro inhibitory assessment of phage SW16-7. (**A**) The growth curves (OD600 nm) of host strains SA161 were compared between the control group and treatment group. The treatment group was mixed with 200 μL of SA161 and SW16-7 at an MOI of 0.1, while the control group received 200 μL of SA161 and an equal volume of LB liquid as a substitute for SW16-7. Error bars, s.d. (n = 3 biological replicate) (**B**) The colony count of host strains SA161 recorded at 4-h intervals over a period of 24 h in both the treatment and control groups. Both treatment and control groups were cultured to the exponential phase with 20 mL each. The treatment group received phage SW16-7 at an MOI of 0.1, while the control group received an equal volume of LB liquid. Error bars, s.d. (n = 3 biological replicate), two-tailed unpaired Student’s *t*-test. In the above, asterisks mark a statistically significant difference (**** *p* < 0.0001). The absence of asterisks indicates that there is no statistical significance.

**Figure 4 microorganisms-11-02090-f004:**
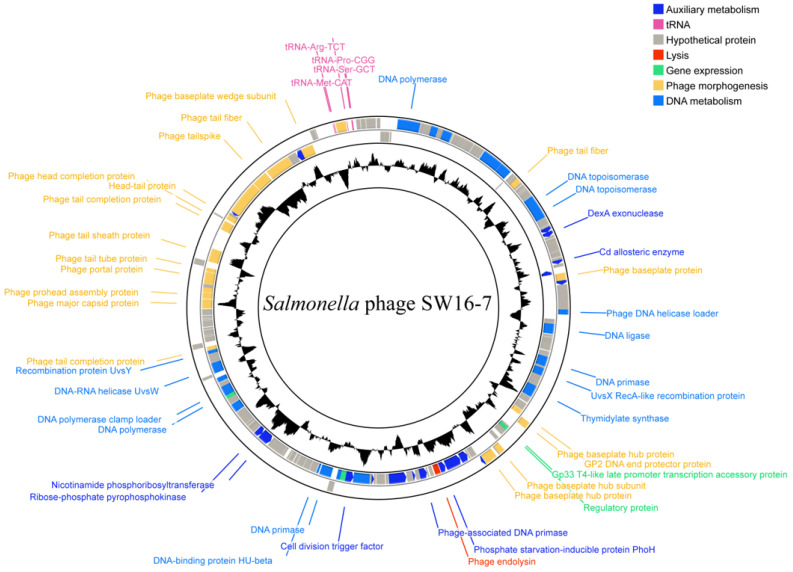
Genome map of SW16-7. The coding domain sequences (CDSs) are labeled by specific colors according to their functional categories. The black circular hologram indicates the GC content. Red: lysis-related gene. Dark blue: auxiliary metabolism related genes. Blue: DNA metabolism related genes. Yellow: phage morphogenesis related genes. Gray: hypothetical protein gene. Green: gene expression related genes. Pink: tRNA.

**Figure 5 microorganisms-11-02090-f005:**
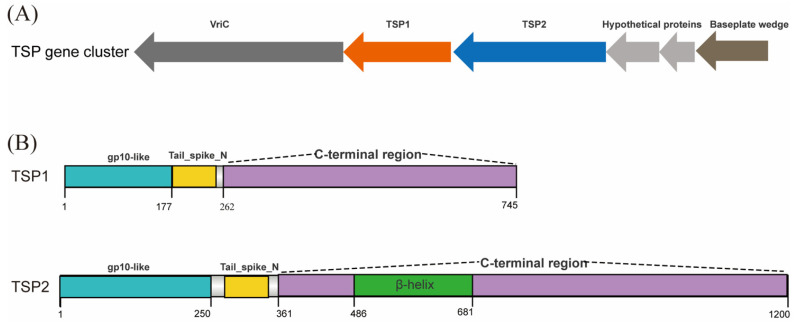
(**A**) The TSP gene cluster of SW16-7. TSP1: Orange, TSP2: Blue (**B**) The domain architecture of TSP1 and TSP2.

**Figure 6 microorganisms-11-02090-f006:**
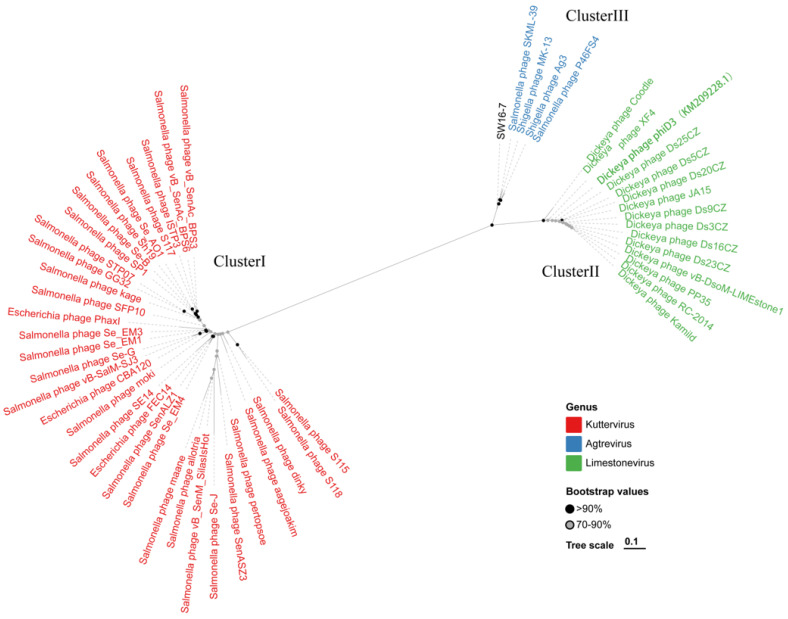
The maximum-likelihood tree constructed by SW16-7 and the 54 other phages within the *Agtrevirus*, *Limestonevirus*, and *Kuttervirus* genus based on the whole genome sequencing. Cluster I contains all the sequences marked in red, belonging to the *Kuttervirus* genus. Cluster II contains all the sequences marked in green, belonging to the *Limestonevirus genus*. Cluster III contains all the sequences marked in blue, belonging to the *Agtrevirus* genus. The bootstrap analysis was performed with 1000 replications. Black dots on branches indicate strong bootstrap support (>90%); grey dots on branches indicate strong bootstrap support (70–90%).

**Figure 7 microorganisms-11-02090-f007:**
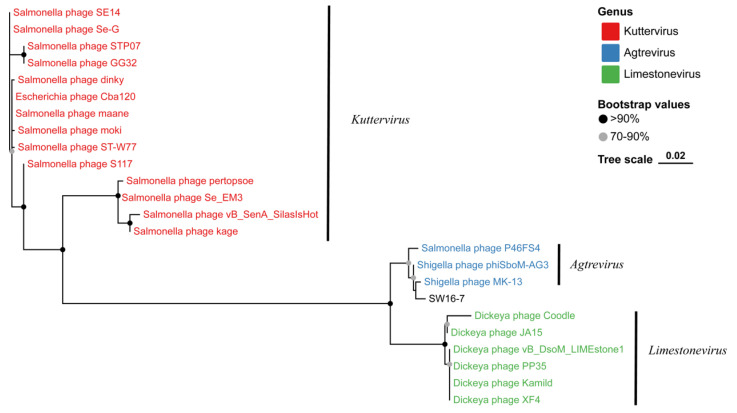
The maximum-likelihood tree of SW16-7 terminase large subunit. The bootstrap analysis was performed with 1000 replications. Numbers above branches are bootstrap values. The bootstrap analysis was performed with 1000 replications. Black dots on branches indicate strong bootstrap support (>90%); grey dots on branches indicate strong bootstrap support (70–90%).

**Table 1 microorganisms-11-02090-t001:** Calculations of average nucleotide identity (ANI) between SW16-7 and other Ackermannviridae phage.

Phages	Genus	OrthoANI
*Salmonella* phage P46FS4	*Agtrevirus*	96.04
*Shigella* phage MK-13	*Agtrevirus*	95.96
*Shigella* phage Ag3	*Agtrevirus*	95.84
*Salmonella* phage SKML-39	*Agtrevirus*	95.57
*Dickeya* phage vB-DsoM-LIMEstone1	*Limestonevirus*	88.24
*Dickeya* phage Ds23CZ	*Limestonevirus*	88.21
*Dickeya* phage Ds3CZ	*Limestonevirus*	88.16
*Dickeya* phage Ds5CZ	*Limestonevirus*	88.16
*Dickeya* phage PP35	*Limestonevirus*	88.12
*Dickeya* phage Ds9CZ	*Limestonevirus*	88.11
*Dickeya* phage Ds20CZ	*Limestonevirus*	88.10
*Dickeya* phage XF4	*Limestonevirus*	88.04
*Dickeya* phage Kamild	*Limestonevirus*	88.04
*Dickeya* phage JA15	*Limestonevirus*	88.02
*Dickeya* phage Ds16CZ	*Limestonevirus*	88.01
*Dickeya* phage RC-2014	*Limestonevirus*	87.86
*Dickeya* phage Ds25CZ	*Limestonevirus*	87.63
*Dickeya* phage Coodle	*Limestonevirus*	87.16
*Salmonella* phage SFP10	*Kuttervirus*	78.91
*Salmonella* phage Sh19	*Kuttervirus*	78.43
*Escherichia* phage vB_EcoA_4HA11	*Kuttervirus*	76.97
*Salmonella* phage pertopsoe	*Kuttervirus*	76.65
*Salmonella* phage allotria	*Kuttervirus*	76.52
*Salmonella* phage vB_SenM_SilasIsHot	*Kuttervirus*	76.44
*Salmonella* phage maane	*Kuttervirus*	76.22
*Salmonella* phage aagejoakim	*Kuttervirus*	75.47
*Salmonella* phage Se_EM4	*Kuttervirus*	75.42
*Salmonella* phage S115	*Kuttervirus*	75.23
*Salmonella* phage SenASZ3	*Kuttervirus*	75.14
*Salmonella* phage Se_AO1	*Kuttervirus*	75.13
*Salmonella* phage SE14	*Kuttervirus*	75.06
*Salmonella* phage Se-J	*Kuttervirus*	75.05
*Salmonella* phage S118	*Kuttervirus*	75.01
*Salmonella* phage STP07	*Kuttervirus*	75.00
*Salmonella* phage dinky	*Kuttervirus*	74.98
*Escherichi*a phage FEC14	*Kuttervirus*	74.98
*Salmonella* phage kage	*Kuttervirus*	74.96
*Salmonella* phage GG32	*Kuttervirus*	74.96
*Salmonella* phage S117	*Kuttervirus*	74.90
*Salmonella* phage Se_EM1	*Kuttervirus*	74.86
*Salmonella* phage vB_SenAc_BPS6	*Kuttervirus*	74.78
*Salmonella* phage vB-SalM-SJ3	*Kuttervirus*	74.77
*Escherichia* phage PhaxI	*Kuttervirus*	74.76
*Salmonella* phage SenALZ1	*Kuttervirus*	74.74
*Escherichia* phage CBA120	*Kuttervirus*	74.68
*Salmonella* phage SP1	*Kuttervirus*	74.67
*Salmonella* phage vB_SenAc_BPS3	*Kuttervirus*	74.63
*Salmonella* phage Se-G	*Kuttervirus*	74.61
*Salmonella* phage ST-W77	*Kuttervirus*	74.59
*Salmonella* phage ISTP3	*Kuttervirus*	74.56
*Salmonella* phage Se-B	*Kuttervirus*	74.55
*Salmonella* phage moki	*Kuttervirus*	74.54
*Salmonella* phage Se_EM3	*Kuttervirus*	74.47

**Table 2 microorganisms-11-02090-t002:** The host range of phage SW16-7.

Salmonella Serovar	Group	No. of Strains Tested	No. of Lysed Strains
*S. Weltevreden*	Group O:3,10	11	11
*S. Muenster*	Group O:3,10	11	11
*S. London*	Group O:3,10	11	9
*S. Meleagridis*	Group O:3,10	7	6
*S. Give*	Group O:3,10	6	5
*S. Anatum*	Group O:3,10	5	4
*S. Derby*	Group O:4	17	2
*S. Stanley*	Group O:4	16	1
*S. Typhimurium*	Group O:4	16	1
*S. Enteritidis*	Group O:9	15	0
*S. Rissen*	Group O:7	16	1
*S. Thompson*	Group O:7	3	1
*S. Goldcoast*	Groups O:6,8	14	2
*S. Hadar*	Groups O:6,8	2	1
*S. Corvallis*	Groups O:8	16	1
*S. Kentucky*	Groups O:8	15	1
*S. Hillingdon*	Group O:9,46	1	1
*S. Gateshead*	Group O:9,46	1	1
*S. Michigan*	Group O:17	1	1

## Data Availability

The genome sequence was deposited in the GenBank under the accession Number OR228402. The raw data supporting the conclusions of this article will be made available by the authors, without undue reservation.

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
