# Peer review of "SW16-7, a Novel Ackermannviridae Bacteriophage with Highly Effective Lytic Activity Targets Salmonella enterica Serovar Weltevreden"

_microorganisms, 2023, doi:10.3390/microorganisms11082090_

Round 1
Reviewer 1 Report
- I was unable to find a genome sequence at Genbank in order to provide specific evidence, but based on similarity of SW16-7 to other sequenced phages, as shown in your Figure 6, your phage’s DNA very likely contains 30 to 50% of thymidines replaced by 5-aminoethoxymethyl-2’-deoxyuridine (5-NeOmdU). See https://doi.org/10.1093/nar/gkab781 for more information on the genetics and biosynthesis of this DNA modification. The key genes in SW16-7 genome will contain matches to Pfam domains thymidylate synthase, aGPT-Pplase1, aGPT-Pplase2, and a P-loop kinase. Hypermodified thymidines almost certainly impact the biology of the virus:host relationship because they block many restriction endonucleases that are part of the host’s genome defense.
- You observed that SW16-7 contains two tailspike homologs related to both Salmonella phages epsilon15 AND epsilon34. In table two, you show host range and that SW16-7 can infect majority of S. enterica subsp. Anatum (which is the host for epsilon15), but it would be interesting to know if they can similarly infect S. enterica subsp. Newington (which is the host for epsilon34). One of the serotype subgroups in your table might already match S. Newington, so all that would be needed is to make that connection. As an aside, one of the things that is so interesting about Salmonella phages epsilon15 and epsilon34 is that lysogeny of S. Anatum by epsilon15 causes lysogenic conversion by expression of phage encoded genes (in the lysogenic state) that modify the O-antigen (gtrAB), such that S. anatum then becomes S. Newington, and thus, sensitive to infection by epsilon34! For more information about this fascinating system, please see the work of Robert Villafane and references therein: http://www.biomedcentral.com/1471-2180/8/227
Reviewer 2 Report
There are a few comments on the text.
1)As presented, the position of Figure 2 with its figure legend on a different page is not ideal. The text on Figure 3 (lines 207-213) could be moved to page 6 above Figure 3 which would allow the Figure 2 legend to be accommodated on page 5 beneath Figure 2 itself. It would be best ensure all figure and table legends are on the same page and the figures and tables themselves.
2)Is line 539 not redundant as it repeats the information in line 538?
3)There are some typographical errors which need to be addressed eg
line 109 ex-periment, delete hyphen
line 260 con-tent, delete hyphen
lines 455, 456, 457, 484, 485 (x2), 487, bootstrap not 'boostrap'
line 684 ...ofSalmonella... add space after 'of '
4) Table 2: Because the numbers of strains tested are small (<17) the fourth column could reasonably be shortened to 'No.of lysed strains'. Including a % value does not add much as the numbers tested are low and certainly not to two decimal places.
Reviewer 3 Report
The authors did extensive characterization of a lytic phage that targets S. Weltevreden and can be a useful biocontrol agent. While the molecular characterization is very detailed, I miss some details about the phenotypical characterization especially as far as statistics are concerned (see below).
L34: "global spread"
L55: "limited activity against"
L83: please state more precisely: most likely you used copper grids filmed with something before they were carbon-coated.
L85: supply information about the TEM and TEM-camera used. How were the grids evaluated? How many phages were measured (as basis for mean and SD in the result section)?
L96: Did you do technical or biological replicates?
L123-127 first experiment: why did you add more salmonella to the control?
L160: which statistic tests were used? Or explain why no statistical testing was done.
L204: If you claim a significant decrease, there should be statistical testing.
Figs. 2 and 3 Standard deviations are minimal or non existent. Did you do technical or biological replicates? In case that technical replicates were done, 2 additional replicates (on different days) should be done to replace the technical replicates.
Fig. 2A: MOI between 1 and 10-6 result in very similar titers. How can you be sure that the minimal increase at MOI 0.1 has any significance? And that this is the optimal titer?
L536-L540: one of these sentences is redundant.
Round 2
Reviewer 1 Report
Thank you for responding to my comments. I think the work is improved as a result. However, you may have misunderstood the second point I was trying to make regarding serovars and phage sensitivity (possibly because I was not very clear about what I was asking for).
Phage SW16-7 has a tailspike closely resembling that of epsilon15 and another tailspike resembling that of epsilon34. Therefore, it stands to reason that SW16-7 should be able to infect the hosts of both of these phages: S. anatum (which you demonstrated), as well as S. newington (which was not listed among the strains in Table 2). So at the time of my initial review, I was wondering if one or more of the strains listed in Table 2 might share the same serotype as S. newington. This would have provided additional support for matching each of the tailspike identities with the serotypes sensitive to the phage. In the meantime, I found a work by L. Le Minor (https://doi.org/10.1007/BF01963091) that indicates S. newington as having an O3:15 serotype. But, unfortunately, none of the strains in Table 2 has that serotype. Had there been a strain in Table 2 that does have the O3:15 serotype, it would have been worth noting that. However, it's not the case, and this is a minor shortcoming which shouldn't hold back the work.
Reviewer 3 Report
The authors have integrated all comments and suggestions.
Author Response
We appreciate your approval of the revised content.